# Kernel Normalized Convolutional Networks

**Reza Nasirigerdeh**                                          *reza.nasirigerdeh@tum.com*
*Technical University of Munich*
*Helmholtz Munich*

**Reihaneh Torkzadehmahani**                      *reihaneh.torkzadehmahani@tum.de*
*Technical University of Munich*

**Daniel Rueckert**                                          *daniel.rueckert@tum.de*
*Technical University of Munich*
*Imperial College London*

**Georgios Kaissis**                                          *g.kaissis@tum.de*
*Technical University of Munich*
*Helmholtz Munich*

**Reviewed on OpenReview:** *https://openreview.net/forum?id=Uv3XVAEgG6*

## Abstract

Existing convolutional neural network architectures frequently rely upon batch normalization (BatchNorm) to effectively train the model. BatchNorm, however, performs poorly with small batch sizes, and is inapplicable to differential privacy. To address these limitations, we propose the **kernel normalization** (**KernelNorm**) and **kernel normalized convolutional** layers, and incorporate them into kernel normalized convolutional networks (**KNConvNets**) as the main building blocks. We implement KNConvNets corresponding to the state-of-the-art ResNets while forgoing the BatchNorm layers. Through extensive experiments, we illustrate that KNConvNets achieve higher or competitive performance compared to the BatchNorm counterparts in image classification and semantic segmentation. They also significantly outperform their batch-independent competitors including those based on layer and group normalization in non-private and differentially private training. Given that, KernelNorm combines the batch-independence property of layer and group normalization with the performance advantage of BatchNorm [1].

## 1 Introduction

Convolutional neural networks (CNNs) (LeCun et al., 1989) are standard architectures in computer vision tasks such as image classification (Krizhevsky et al., 2012; Sermanet et al., 2014) and semantic segmentation (Long et al., 2015b). Deep CNNs including ResNets (He et al., 2016a) achieved outstanding performance in classification of challenging datasets such as ImageNet (Deng et al., 2009). One of the main building blocks of these CNNs is *batch normalization* (BatchNorm) (Ioffe & Szegedy, 2015). The BatchNorm layer considerably enhances the performance of deep CNNs by smoothening the optimization landscape (Santurkar et al., 2018), and addressing the problem of vanishing gradients (Bengio et al., 1994; Glorot & Bengio, 2010).

BatchNorm, however, has the disadvantage of breaking the independence among the samples in the batch (Brock et al., 2021b). This is because BatchNorm carries out normalization along the batch dimension (Figure 1a), and as a result, the normalized value associated with a given sample depends on the statistics of the other samples in the batch. Consequently, the effectiveness of BatchNorm is highly dependent on

---

[1]The code is available at: `https://github.com/reza-nasirigerdeh/norm-torch`

batch size. With large batch sizes, the batch normalized models are trained effectively due to more accurate estimation of the batch statistics. Using small batch sizes, on the other hand, BatchNorm causes reduction in model accuracy (Wu & He, 2018) because of dramatic fluctuations in the batch statistics. BatchNorm, moreover, is inapplicable to *differential privacy* (DP) (Dwork & Roth, 2014). For the theoretical guarantees of DP to hold for the training of neural networks (Abadi et al., 2016), it is required to compute the gradients individually for each sample in a batch, clip the per-sample gradients, and then average and inject random noise to limit the information learnt about any particular sample. Because per-sample (individual) gradients are required, the gradients of a given sample are not allowed to be influenced by other samples in the batch. This is not the case for BatchNorm, where samples are normalized using the statistics computed over the other samples in the batch. Consequently, BatchNorm is inherently incompatible with DP.

To overcome the limitations of BatchNorm, the community has introduced *batch-independent* normalization layers including layer normalization (LayerNorm) (Ba et al., 2016), instance normalization (InstanceNorm) (Ulyanov et al., 2016), group normalization (GroupNorm) (Wu & He, 2018), positional normalization (PositionalNorm) (Li et al., 2019), and local context normalization (LocalContextNorm) (Ortiz et al., 2020), which perform normalization independently for each sample in the batch. These layers do not suffer from the drawbacks of BatchNorm, and might outperform BatchNorm in particular domains such as generative tasks (e.g. LayerNorm in Transformer models (Vaswani et al., 2017)). For image classification and semantic segmentation, however, they typically do not achieve performance comparable with BatchNorm's in non-private (without DP) training. In DP, moreover, these batch-independent layers might not provide the accuracy gain we expect compared to non-private learning. This motivates us to develop alternative layers, which are batch-independent but more efficient in both non-private and differentially private learning.

Our main contribution is to propose two novel *batch-independent* layers called **kernel normalization** (**KernelNorm**) and the **kernel normalized convolutional** (**KNConv**) layer to further enhance the performance of deep CNNs. The distinguishing characteristic of the proposed layers is that they *extensively* take into account the *spatial correlation* among the elements during normalization. KernelNorm is similar to a pooling layer, except that it normalizes the elements specified by the kernel window instead of computing the average/maximum of the elements, and it operates over all input channels instead of a single channel (Figure 1g). KNConv is the combination of KernelNorm with a convolutional layer, where it applies KernelNorm to the input, and feeds KernelNorm's output to the convolutional layer (Figure 2). From another perspective, KNConv is the same as the convolutional layer except that KNConv first normalizes the input elements specified by the kernel window, and then computes the convolution between the normalized elements and kernel weights. In both aforementioned naive forms, however, KNConv is computationally inefficient because it leads to extremely large number of normalization units, and therefore, considerable computational overhead to normalize the corresponding elements. To tackle this issue, we present **computationally-efficient KNConv** (Algorithm 1), where the output of the convolution is adjusted using the mean and variance of the normalization units. This way, it is not required to normalize the elements, improving the computation time by orders of magnitude.

As an application of the proposed layers, we introduce **kernel normalized convolutional networks** (**KNConvNets**) corresponding to residual networks (He et al., 2016a), referred to as **KNResNets**, which employ KernelNorm and computationally-efficient KNConv as the main building blocks while forgoing the BatchNorm layers (Section 3). Our last contribution is to draw performance comparisons among KNResNets and the competitors using several benchmark datasets including CIFAR-100 (Krizhevsky et al., 2009), ImageNet (Deng et al., 2009), and Cityscapes (Cordts et al., 2016). According to the experimental results (Section 4), KNResNets deliver significantly higher accuracy than the BatchNorm counterparts in image classification on CIFAR-100 using a small batch size. KNResNets, moreover, achieve higher or competitive performance compared to the batch normalized ResNets in classification on ImageNet and semantic segmentation on CityScapes. Furthermore, KNResNets considerably outperform GroupNorm and LayerNorm based models for almost all considered case studies in non-private and differentially private learning. Considering that, KernelNorm combines the performance advantage of BatchNorm with the batch-independence benefit of LayerNorm and GroupNorm.

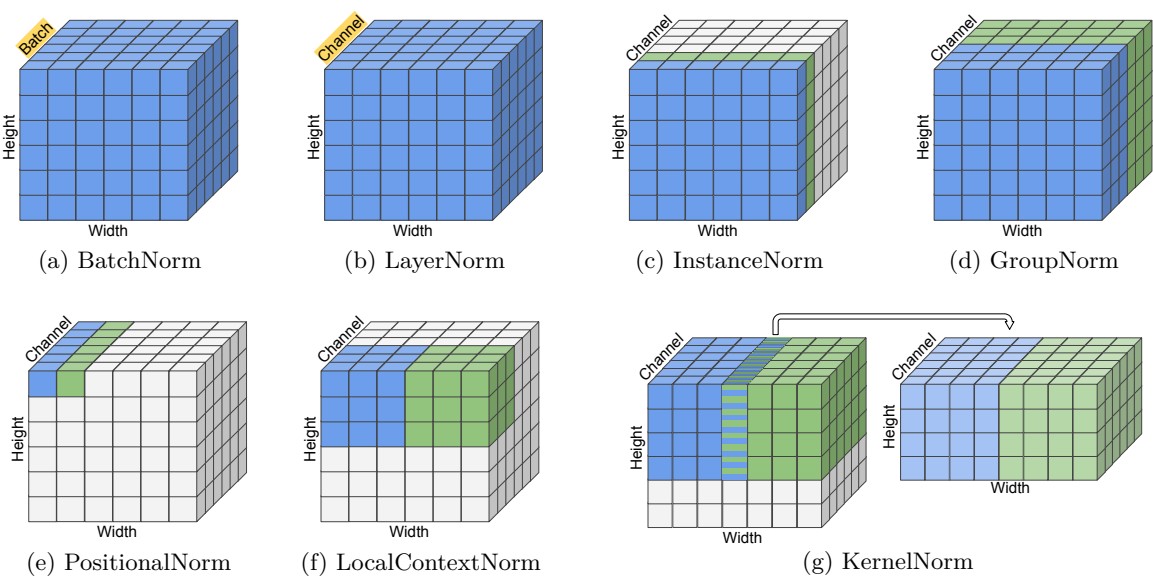

Figure 1: **Normalization layers** differ from one another in their normalization unit (highlighted in blue and green). The normalization layers in (a)-(f) establish a *one-to-one correspondence* between the input and normalized elements (i.e. no overlap between the normalization units, and no ignorance of an element). The proposed **KernelNorm** layer does not impose such one-to-one correspondence: Some elements (dash-hatched area) are common among the normalization units, contributing more than once to the output, while some elements (uncolored ones) are ignored during normalization. Due to this unique property of overlapping normalization units, KernelNorm *extensively* incorporates the spatial correlation among the elements during normalization (akin to the convolutional layer), which is not the case for the other normalization layers.

## 2   Normalization Layers

Normalization methods can be categorized into *input normalization* and *weight normalization* (Salimans & Kingma, 2016; Bansal et al., 2018; Wang et al., 2020; Qi et al., 2020). The former techniques perform normalization on the input tensor, while the latter ones normalize the model weights. The aforementioned layers including BatchNorm, and the proposed KernelNorm layer as well as *divisive normalization* (Heeger, 1992; Bonds, 1989), (Ren et al., 2017) and *local response normalization* (LocalResponseNorm) (Krizhevsky et al., 2012) belong to the category of input normalization. Weight standardization (Huang et al., 2017b; Qiao et al., 2019) and normalizer-free networks (Brock et al., 2021a) fall into the category of weight normalization.

In the following, we provide an overview on the existing normalization layers closely related to KernelNorm, i.e. the layers which are based on input normalization, and employ standard normalization (zero-mean and unit-variance) to normalize the input tensor. For the sake of simplicity, we focus on 2D images, but the concepts are also applicable to 3D images. For a 2D image, the input of a layer is a 4D tensor of shape ($n$, $c$, $h$, $w$), where $n$ is batch size, $c$ is the number of input channels, $h$ is height, and $w$ is width of the tensor. Normalization layers differ from one another in their *normalization unit*, which is a group of input elements that are normalized together with the mean and variance of the unit.

The normalization unit of **BatchNorm** (Figure 1a) is a 3D tensor of shape ($n$, $h$, $w$), implying that BatchNorm incorporates all elements in the batch, height, and width dimensions during normalization. **LayerNorm**'s normalization unit (Figure 1b) is a 3D tensor of shape ($c$, $h$, $w$), i.e. LayerNorm considers all elements in the channel, height, and width dimensions for normalization. The normalization unit of **InstanceNorm** (Figure 1c) is a 2D tensor of shape ($h$, $w$), i.e. all elements of the height and width dimensions are taken into account during normalization.

**GroupNorm**'s normalization unit (Figure 1d) is a 3D tensor of shape ($c_g$, $h$, $w$), where $c_g$ indicates the channel group size. Thus, GroupNorm incorporates all elements in the height and width dimensions and a

subset of elements specified by the group size in the channel dimension during normalization. **Positional-Norm**'s normalization unit (Figure 1e) is a 1D tensor of shape $c$, i.e. PositionalNorm performs channel-wise normalization. The normalization unit of **LocalContextNorm** (Figure 1f) is a 3D tensor of shape ($c_g$, $r$, $s$), where $c_g$ is the group size, and ($r$, $s$) is the window size. Therefore, LocalContextNorm considers a subset of elements in the height, width, and channel dimensions during normalization.

BatchNorm, LayerNorm, InstanceNorm, and GroupNorm consider *all elements* in the height and width dimensions for normalization, and thus, they are referred to as *global normalization* layers. PositionalNorm and LocalContextNorm, on the other hand, are called *local normalization* layers (Ortiz et al., 2020) because they incorporate a *subset of elements* from the aforementioned dimensions during normalization. In spite of their differences, the aforementioned normalization layers including BatchNorm have at least one thing in common: There is a *one-to-one correspondence* between the original elements in the input and the normalized elements in the output. That is, there is exactly one normalized element associated with each input element. Therefore, these layers do not modify the shape of the input during normalization.

## 3 Kernel Normalized Convolutional Networks

The KernelNorm and KNConv layers are the main building blocks of KNConvNets. **KernelNorm** takes the kernel size ($k_h$, $k_w$), stride ($s_h$, $s_w$), padding ($p_h$, $p_w$), and dropout probability $p$ as hyper-parameters. It pads the input with zeros if padding is specified. The normalization unit of KernelNorm (Figure 1g) is a tensor of shape ($c$, $k_h$, $k_w$), i.e. KernelNorm incorporates *all elements* in the channel dimension but a *subset of elements* specified by the kernel size from the height and width dimensions during normalization. The KernelNorm layer (1) applies random dropout (Srivastava et al., 2014) to the normalization unit to obtain the *dropped-out* unit, (2) computes mean and variance of the dropped-out unit, and (3) employs the calculated mean and variance to normalize the *original* normalization unit:

$$U' = D_p(U), \tag{1}$$

$$\mu_{u'} = \frac{1}{c \cdot k_h \cdot k_w} \cdot \sum_{i_c=1}^{c} \sum_{i_h=1}^{k_h} \sum_{i_w=1}^{k_w} U'(i_c, i_h, i_w),$$

$$\sigma_{u'}^2 = \frac{1}{c \cdot k_h \cdot k_w} \cdot \sum_{i_c=1}^{c} \sum_{i_h=1}^{k_h} \sum_{i_w=1}^{k_w} (U'(i_c, i_h, i_w) - \mu_{u'})^2, \tag{2}$$

$$\hat{U} = \frac{U - \mu_{u'}}{\sqrt{\sigma_{u'}^2 + \epsilon}}, \tag{3}$$

where $p$ is the dropout probability, $D_p$ is the dropout operation, $U$ is the normalization unit, $U'$ is the dropped-out unit, $\mu_{u'}$ and $\sigma_{u'}^2$ are the mean and variance of the dropped-out unit, respectively, $\epsilon$ is a small number (e.g. $10^{-5}$) for numerical stability, and $\hat{U}$ is the normalized unit.

Partially inspired by BatchNorm, KernelNorm introduces a *regularizing effect* during training by intentionally normalizing the elements of the original unit $U$ using the statistics computed over the dropped-out unit $U'$. In BatchNorm, the normalization statistics are computed over the batch but not the whole dataset, where the mean and variance of the batch are randomized approximations of those from the whole dataset. The "stochasticity from the batch statistics" creates a regularizing effect in BatchNorm according to Ba et al. (2016). KernelNorm employs dropout to generate similar stochasticity in the mean and variance of the normalization unit. Notice that the naive option of injecting random noise directly into the mean and variance might generate too much randomness, and hinder model convergence. Using dropout in the aforementioned fashion, KernelNorm can control the regularization effect with more flexibility.

The first normalization unit of KernelNorm is bounded to a window specified by diagonal points $(1, 1)$ and $(k_h, k_w)$ in the height and width dimensions. The coordinates of the next normalization unit are $(1, 1 + s_w)$ and $(k_h, k_w + s_w)$, which are obtained by sliding the window $s_w$ elements along the width dimension. If there are not enough elements for kernel in the width dimension, the window is slid by $s_h$ elements in the height dimension, and the above procedure is repeated. Notice that KernelNorm works on the padded input of shape ($n$, $c$, $h + 2 \cdot p_h$, $w + 2 \cdot p_w$), where ($p_h$, $p_w$) is the padding size. The output $\hat{X}$ of KernelNorm

is the concatenation of the *normalized units* $\hat{U}$ from Equation 3 along the height and width dimensions. KernelNorm's output is of shape $(n, c, h_{out}, w_{out})$, and it has total of $n \cdot \frac{h_{out}}{k_h} \cdot \frac{w_{out}}{k_w}$ normalization units, where $h_{out}$ and $w_{out}$ are computed as follows:

$$h_{out} = k_h \cdot \lfloor \frac{h + 2 \cdot p_h - k_h}{s_h} + 1 \rfloor, w_{out} = k_w \cdot \lfloor \frac{w + 2 \cdot p_w - k_w}{s_w} + 1 \rfloor$$

In simple terms, KernelNorm behaves similarly to the pooling layers with two major differences: (1) KernelNorm normalizes the elements specified by the kernel size instead of computing the maximum/average over the elements, and (2) KernelNorm operates over all channels rather than a single channel. KernelNorm is a *batch-independent* and *local normalization* layer, but differs from the existing normalization layers in two aspects: (I) There is not necessarily a one-to-one correspondence between the original elements in the input and the normalized elements in the output of KernelNorm. Stride values less than kernel size lead to overlapping normalization units, where some input elements contribute more than once in the output (akin to the convolutional layer). If the stride value is greater than kernel size, some input elements are completely ignored during normalization. Therefore, the output shape of KernelNorm can be different from the input shape. (II) KernelNorm can *extensively* take into account the *spatial correlation* among the elements during normalization because of the overlapping normalization units.

**KNConv** is the combination of KernelNorm and the traditional convolutional layer (Figure 2). It takes the number of input channels $ch_{in}$, number of output channels (filters) $ch_{out}$, kernel size $(k_h, k_w)$, stride $(s_h, s_w)$, and padding $(p_h, p_w)$, exactly the same as the convolutional layer, as well as the dropout probability $p$ as hyper-parameters. KNConv first applies KernelNorm with kernel size $(k_h, k_w)$, stride $(s_h, s_w)$, padding $(p_h, p_w)$, and dropout probability $p$ to the input tensor. Next, it applies the convolutional layer with $ch_{in}$ channels, $ch_{out}$ filters, kernel size $(k_h, k_w)$, stride $(k_h, k_w)$, and padding of zero to the output of KernelNorm. That is, both kernel size and stride values of the convolutional layer are identical to kernel size of KernelNorm.

From another perspective, KNConv is the same as the convolutional layer except that it normalizes the input elements specified by the kernel window before computing the convolution. Assuming that $U$ contains the input elements specified by the kernel window, $\hat{U}$ is the normalized version of $U$ from KernelNorm (Equation 3), $Z$ is the kernel weights of a given filter, $\star$ is the convolution (or dot product) operation, and $b$ is the bias value, KNConv computes the output as follows:

$$\text{KNConv}(U, Z, b) = \hat{U} \star Z + b \tag{4}$$

KNConv (or in fact KernelNorm) leads to extremely high number of normalization units, and consequently, remarkable computational overhead. Thus, KNConv in its simple format outlined in Equation 4 (or as a combination of the KernelNorm and convolutional layers) is computationally inefficient. Compared to the convolutional layer, the additional computational overhead of KNConv originates from (I) calculating the mean and variance of the units using Equation 2, and (II) normalizing the elements by the mean and variance using Equation 3.

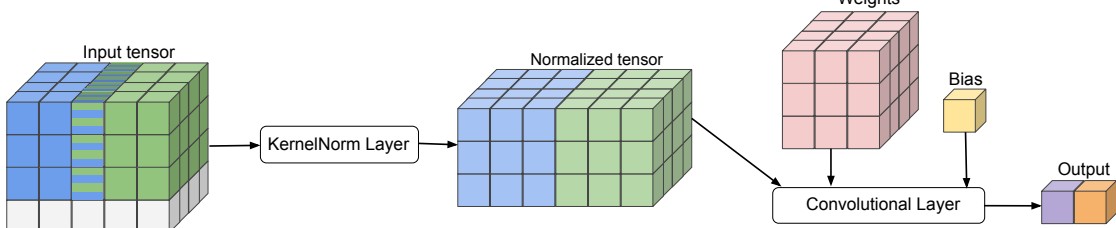

Figure 2: **KNConv** as the combination of the KernelNorm and convolutional layers. KNConv first applies KernelNorm with kernel size $(3, 3)$ and stride $(2,2)$ to the input tensor, and then gives KernelNorm's output to a convolutional layer with kernel size and stride $(3, 3)$. That is, the kernel size and stride of the convolutional layer and the kernel size of KernelNorm are identical.

**Computationally-efficient KNConv** reformulates Equation 4 in a way that it completely eliminates the overhead of normalizing the elements:

$$\text{KNConv}(U, Z, b) = \hat{U} \star Z + b = \sum_{i_c=1}^{c} \sum_{i_h=1}^{k_h} \sum_{i_w=1}^{k_w} (\frac{U(i_c, i_h, i_w) - \mu_{u'}}{\sqrt{\sigma_{u'}^2 + \epsilon}}) \cdot Z(i_c, i_h, i_w) + b$$

$$= (\sum_{i_c=1}^{c} \sum_{i_h=1}^{k_h} \sum_{i_w=1}^{k_w} U(i_c, i_h, i_w) \cdot Z(i_c, i_h, i_w) - \mu_{u'} \cdot \sum_{i_c=1}^{c} \sum_{i_h=1}^{k_h} \sum_{i_w=1}^{k_w} Z(i_c, i_h, i_w)) \cdot \frac{1}{\sqrt{\sigma_{u'}^2 + \epsilon}} + b \quad (5)$$

$$= (U \star Z - \mu_{u'} \cdot \sum_{i_c=1}^{c} \sum_{i_h=1}^{k_h} \sum_{i_w=1}^{k_w} Z(i_c, i_h, i_w)) \cdot \frac{1}{\sqrt{\sigma_{u'}^2 + \epsilon}} + b$$

According to Equation 5 and Algorithm 1, KNConv applies the convolutional layer to the original unit, computes the mean and standard deviation of the dropped-out unit as well as the sum of the kernel weights, and finally adjusts the convolution output using the computed statistics. This way, it is not required to normalize the elements, improving the computation time of KNConv by orders of magnitude.

In terms of implementation, KernelNorm employs the *unfolding* operation in PyTorch (2023b) to implement the sliding window mechanism in the *kn_mean_var* function in Algorithm 1. Moreover, it uses the *var_mean* function in PyTorch (2023c) to compute the mean and variance over the unfolded tensor along the channel, width, and height dimensions.

The defining characteristic of KernelNorm and KNConv is that they take into consideration the *spatial correlation* among the elements during normalization on condition that the kernel size is greater than 1×1. Existing architectures (initially designed for global normalization), however, do not satisfy this condition. For instance, all ResNets use 1×1 convolution for downsampling and increasing the number of filters. ResNet-50/101/152, in particular, contains bottleneck blocks with a single 3×3 and two 1×1 convolutional layers. Consequently, the current architectures are unable to fully utilize the potential of kernel normalization.

**KNConvNets** are bespoke architectures for kernel normalization, consisting of computationally-efficient KNConv and KernelNorm as the main building blocks. KNConvNets are *batch-independent* (free of Batch-Norm), which primarily employ kernel sizes of 2×2 or 3×3 to benefit from the spatial correlation of elements during normalization. In this study, we propose KNConvNets corresponding to ResNets, called *KNResNets*, for image classification and semantic segmentation.

---

**Algorithm 1:** Computationally-efficient KNConv layer

**Input:** input tensor $X$, number of input channels $ch_{in}$, number of output channels $ch_{out}$, kernel size $(k_h, k_w)$, stride $(s_h, s_w)$, padding $(p_h, p_w)$, *bias* flag, dropout probability $p$, and epsilon $\epsilon$

`// 2-dimensional convolutional layer`
conv_layer = Conv2d(in_channels=$ch_{in}$, out_channels=$ch_{out}$, kernel_size=$(k_h, k_w)$, stride=$(s_h, s_w)$, padding=$(p_h, p_w)$, bias=$false$)
`// convolutional layer output`
conv_out = conv_layer(input=$X$)
`// mean and variance from KernelNorm`
$\mu$, $\sigma^2$ = kn_mean_var(input=$X$, kernel_size=$(k_h, k_w)$, stride=$(s_h, s_w)$, padding=$(p_h, p_w)$, dropout_p=$p$)
`// KNConv output`
kn_conv_out = (conv_out - $\mu \cdot \sum$ conv_layer.weights) / $\sqrt{\sigma^2 + \epsilon}$
`// apply bias`
**if** bias **then**
   kn_conv_out += conv_layer.bias

**Output:** kn_conv_out

---

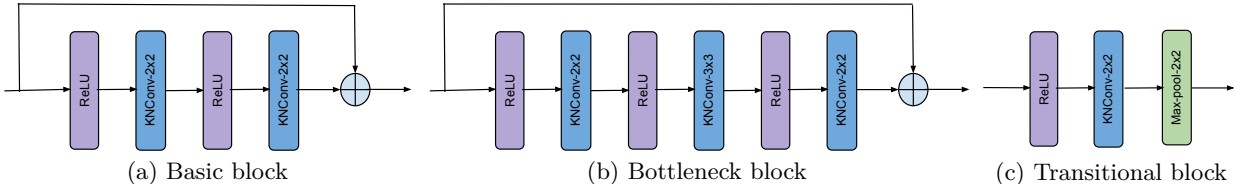

Figure 3: **KNResNet blocks**: Basic blocks are employed in KNResNet-18/34, while KNResNet-50 is based on bottleneck blocks. Transitional blocks are used in all KNResNets for increasing the number of filters and downsampling. The architectures of KNResNet-18/34/50 are available in Figures 5-6 in Appendix A.

**KNResNets** comprise three types of blocks: residual *basic* block, residual *bottleneck* block, and *transitional* block (Figure 3). Basic blocks contain two KNConv layers with kernel size of 2×2, whereas bottleneck blocks consist of three KNConv layers with kernel sizes of 2×2, 3×3, and 2×2, respectively. The stride value in both basic and bottleneck blocks is 1×1. The padding values of the first and last KNConv layers, however, are 1×1 and zero so that the width and height of the output remain identical to the input's (necessary condition for residual blocks with identity shortcut). The middle KNConv layer in bottleneck blocks uses 1×1 padding. Transitional blocks include a KNConv layer with kernel size of 2×2 and stride of 1×1 to increase the number of filters, and a max-pooling layer with kernel size and stride of 2×2 to downsample the input.

We propose the KNResNet-18, KNResNet-34, and KNResNet-50 architectures based on the aforementioned block types (Figure 5 in Appendix A). KNResNet-18/34 uses basic and transitional blocks, while KNResNet-50 mainly employs bottleneck and transitional blocks. For semantic segmentation, we utilize KNResNet-18/34/50 as backbone (Figure 6 in Appendix A), but the kernel size of the KNConv and max-pooling layers in basic and transitional blocks is 3×3 instead of 2×2.

## 4 Evaluation

We compare the performance of KNResNets to the BatchNorm, GroupNorm, LayerNorm, and LocalContextNorm counterparts. For image classification, we do not include LocalContextNorm in our evaluation because its performance is similar to GroupNorm (Ortiz et al., 2020). The experimental evaluation is divided into four categories: (I) batch size-dependent performance analysis, (II) image classification on ImageNet, (III) semantic segmentation on Cityscapes, and (IV) differentially private image classification on ImageNet32×32.

We adopt the original implementation of ResNet-18/34/50 from PyTorch (Paszke et al., 2019), and the PreactResNet-18/34/50 (He et al., 2016b) implementation from Kuang (2021). The architectures are based on BatchNorm. For GroupNorm/LocalContextNorm related models, BatchNorm is replaced by Group-Norm/LocalContextNorm. Regarding LayerNorm based architectures, GroupNorm with number of groups of 1 (equivalent to LayerNorm) is substituted for BatchNorm. The number of groups of GroupNorm is 32 (Wu & He, 2018). The number of groups and window size for LocalContextNorm are 2 and 227×227, respectively (Ortiz et al., 2020).

For low-resolution datasets (CIFAR-100 and ImageNet32×32), we replace the first 7×7 convolutional layer with a 3×3 convolutional layer and remove the following max-pooling layer. Moreover, we insert a normalization layer followed by an activation function before the last average-pooling layer in the PreactResNet architectures akin to KNResNets (Figure 5 at Appendix A). The aforementioned modifications considerably enhance the accuracy of the competitors. For semantic segmentation, we employ the fully convolutional network architecture (Long et al., 2015a) with BatchNorm, GroupNorm, LayerNorm, and LocalContextNorm based ResNet-18/34/50 as backbone. For KNResNets, we use fully convolutional versions of KNResNet-18/34/50 (Figure 6 at Appendix A).

Table 1: Test accuracy versus batch size on **CIFAR-100**.

| Model | Normalization | Parameters | B=2 | B=32 | B=256 |
|---|---|---|---|---|---|
| ResNet-18-LN | LayerNorm | 11.220 M | 72.68±0.22 | 73.17±0.16 | 71.99±0.45 |
| PreactResNet-18-LN | LayerNorm | 11.220 M | 73.51±0.10 | 73.36±0.15 | 72.91±0.07 |
| ResNet-18-GN | GroupNorm | 11.220 M | 74.62±0.12 | 74.46±0.05 | 74.46±0.08 |
| PreactResNet-18-GN | GroupNorm | 11.220 M | 74.82±0.24 | 74.74±0.44 | 74.62±0.36 |
| ResNet-18-BN | BatchNorm | 11.220 M | 72.11±0.25 | 78.52±0.20 | 77.72±0.04 |
| PreactResNet-18-BN | BatchNorm | 11.220 M | 72.57±0.19 | 78.32±0.09 | 77.83±0.16 |
| KNResNet-18 (ours) | KernelNorm | 11.216 M | **79.10**±0.10 | **79.29**±0.02 | **78.84**±0.10 |
| ResNet-34-LN | LayerNorm | 21.328 M | 73.74±0.26 | 73.88±0.37 | 72.48±0.57 |
| PreactResNet-34-LN | LayerNorm | 21.328 M | 74.79±0.13 | 74.34±0.42 | 73.10±0.42 |
| ResNet-34-GN | GroupNorm | 21.328 M | 75.76±0.14 | 75.72±0.06 | 75.44±0.27 |
| PreactResNet-34-GN | GroupNorm | 21.328 M | 75.82±0.05 | 75.85±0.28 | 75.76±0.25 |
| ResNet-34-BN | BatchNorm | 21.328 M | 73.06±0.23 | 79.21±0.09 | 78.27±0.19 |
| PreactResNet-34-BN | BatchNorm | 21.328 M | 72.20±0.19 | 79.09±0.03 | 78.59±0.24 |
| KNResNet-34 (ours) | KernelNorm | 21.323 M | **79.28**±0.09 | **79.53**±0.15 | **79.16**±0.21 |
| ResNet-50-LN | LayerNorm | 23.705 M | 75.83±0.25 | 75.74±0.14 | 74.37±0.58 |
| PreactResNet-50-LN | LayerNorm | 23.705 M | 74.28±0.31 | 74.57±0.32 | 73.41±0.15 |
| ResNet-50-GN | GroupNorm | 23.705 M | 77.03±0.62 | 77.02±0.08 | 74.79±0.14 |
| PreactResNet-50-GN | GroupNorm | 23.705 M | 75.67±0.27 | 76.08±0.18 | 75.52±0.13 |
| ResNet-50-BN | BatchNorm | 23.705 M | 71.02±0.15 | **80.39**±0.06 | 77.89±0.06 |
| PreactResNet-50-BN | BatchNorm | 23.705 M | 70.83±0.41 | 80.28±0.15 | 78.88±0.21 |
| KNResNet-50 (ours) | KernelNorm | 23.682 M | **80.24**±0.18 | 80.18±0.10 | **80.09**±0.26 |

## 4.1 Batch size-dependent performance analysis

**Dataset.** The CIFAR-100 dataset consists of 50000 train and 10000 test samples of shape 32×32 from 100 classes. We adopt the data preprocessing and augmentation scheme widely used for the dataset (Huang et al., 2017a; He et al., 2016b;a): Horizontally flipping and randomly cropping the samples after padding them. The cropping and padding sizes are 32×32 and 4×4, respectively. Additionally, the feature values are divided by 255 for KNResNets, whereas they are normalized using the mean and standard deviation (SD) of the dataset for the competitors.

**Training.** The models are trained for 150 epochs using the cosine annealing scheduler (Loshchilov & Hutter, 2017) with learning rate decay of 0.01. The optimizer is SGD with momentum of 0.9 and weight decay of 0.0005. For learning rate tuning, we run a given experiment with initial learning rate of 0.2, divide it by 2, and re-run the experiment. We continue this procedure until finding the best learning rate (Table 5 in Appendix B). Then, we repeat the experiment three times, and report the mean and SD over the runs.

**Results.** Table 1 lists the test accuracy values achieved by the models for different batch sizes. According to the table, (I) KNResNets dramatically outperform the BatchNorm counterparts for batch size of 2, (II) KNResNets deliver highly competitive accuracy values compared to BatchNorm-based models with batch sizes of 32 and 256, and (III) KNResNets achieve significantly higher accuracy than the batch-independent competitors (LayerNorm and GroupNorm) for all considered batch sizes.

## 4.2 Image classification on ImageNet

**Dataset.** The ImageNet dataset contains around 1.28 million training and 50000 validation images. Following the data preprocessing and augmentation scheme from TorchVision (2023a), the train images are horizontally flipped and randomly cropped to 224×224. The test images are first resized to 256×256, and then center cropped to 224×224. The feature values are normalized using the mean and SD of ImageNet.

**Training.** We follow the experimental setting from Wu & He (2018) and use the multi-GPU training script from TorchVision (2023a) to train KNResNets and the competitors. We train all models for 100 epochs with total batch size of 256 (8 GPUs with batch size of 32 per GPU) using learning rate of 0.1, which is divided by 10 at epochs 30, 60, and 90. The optimizer is SGD with momentum of 0.9 and weight decay of 0.0001.

Table 2: Image classification on **ImageNet**.

| Model | Normalization | Parameters | Top-1 accuracy |
|---|---|---|---|
| ResNet-18-LN | LayerNorm | 11.690 M | 68.34 |
| ResNet-18-GN | GroupNorm | 11.690 M | 68.93 |
| ResNet-18-BN | BatchNorm | 11.690 M | 70.28 |
| KNResNet-18 (ours) | KernelNorm | 11.685 M | **71.17** |
| ResNet-34-LN | LayerNorm | 21.798 M | 71.64 |
| ResNet-34-GN | GroupNorm | 21.798 M | 72.63 |
| ResNet-34-BN | BatchNorm | 21.798 M | 73.99 |
| KNResNet-34 (ours) | KernelNorm | 21.793 M | **74.60** |
| ResNet-50-LN | LayerNorm | 25.557 M | 73.80 |
| ResNet-50-GN | GroupNorm | 25.557 M | 75.92 |
| ResNet-50-BN | BatchNorm | 25.557 M | **76.41** |
| KNResNet-50 (ours) | KernelNorm | 25.556 M | **76.54** |

**Results.** Table 2 demonstrates the Top-1 accuracy values on ImageNet for different architectures. As shown in the table, (I) KNResNet-18 and KNResNet-34 outperform the BatchNorm counterparts by around 0.9% and 0.6%, respectively, (II) KNResNet-18/34/50 achieves higher accuracy (by about 0.6%-3.0%) than LayerNorm and GroupNorm based competitors, and (III) KNResNet-50 delivers almost the same accuracy as the batch normalized ResNet-50.

## 4.3 Semantic segmentation on CityScapes

**Dataset.** The CityScapes dataset contains 2975 train and 500 validation images from 30 classes, 19 of which are employed for evaluation. Following Sun et al. (2019); Ortiz et al. (2020), the train samples are randomly cropped from 2048×1024 to 1024×512, horizontally flipped, and randomly scaled in the range of [0.5, 2.0]. The models are tested on the validation images, which are of shape 2048×1024.

**Training.** Following Sun et al. (2019); Ortiz et al. (2020), we train the models with learning rate of 0.01, which is gradually decayed by power of 0.9. The models are trained for 500 epochs using 2 GPUs with batch size of 8 per GPU. The optimizer is SGD with momentum of 0.9 and weight decay of 0.0005. Notice that we use SyncBatchNorm instead of BatchNorm in the batch normalized models.

Table 3: Semantic segmentation on **CityScapes**.

| Model | Normalization | Parameters | mIoU | Pixel accuracy | Mean accuracy |
|---|---|---|---|---|---|
| ResNet-18-LN | LayerNorm | 13.547 M | 59.10±0.46 | 92.42±0.17 | 69.43±0.58 |
| ResNet-18-GN | GroupNorm | 13.547 M | 62.33±0.52 | 93.23±0.01 | 71.58±0.55 |
| ResNet-18-LCN | LocalContextNorm | 13.547 M | 62.25±0.67 | 92.99±0.06 | 71.59±0.68 |
| ResNet-18-BN | BatchNorm | 13.547 M | 63.90±0.06 | **93.77**±0.02 | 73.15±0.14 |
| KNResNet-18 (ours) | KernelNorm | 13.525 M | **64.37**±0.14 | **93.73**±0.01 | **73.46**±0.12 |
| ResNet-34-LN | LayerNorm | 23.655 M | 60.19±0.32 | 92.73±0.17 | 70.12±0.33 |
| ResNet-34-GN | GroupNorm | 23.655 M | 64.21±0.58 | 93.59±0.07 | 74.32±0.49 |
| ResNet-34-LCN | LocalContextNorm | 23.655 M | 64.75±0.38 | 93.31±0.09 | 74.25±0.37 |
| ResNet-34-BN | BatchNorm | 23.655 M | 66.94±0.34 | **94.27**±0.03 | **76.50**±0.41 |
| KNResNet-34 (ours) | KernelNorm | 23.399 M | **67.61**±0.17 | **94.13**±0.05 | **76.58**±0.19 |
| ResNet-50-LN | LayerNorm | 32.955 M | 57.88±0.84 | 92.31±0.21 | 68.25±0.75 |
| ResNet-50-GN | GroupNorm | 32.955 M | 62.14±0.68 | 93.34±0.04 | 71.66±0.64 |
| ResNet-50-LCN | LocalContextNorm | 32.955 M | 64.03±0.02 | 93.07±0.14 | 73.40±0.03 |
| ResNet-50-BN | BatchNorm | 32.955 M | 65.19±0.50 | 93.98±0.03 | 74.65±0.62 |
| KNResNet-50 (ours) | KernelNorm | 32.874 M | **68.02**±0.13 | **94.22**±0.04 | **77.03**±0.05 |

**Results.** Table 3 lists the mean of class-wise intersection over union (mIoU), pixel accuracy, and mean of class-wise pixel accuracy for different architectures. According to the table, (I) KNResNet-18/34 and the BatchNorm-based counterparts achieve highly competitive mIoU, pixel accuracy, and mean accuracy, whereas KNResNet-50 delivers considerably higher mIoU and mean accuracy than batch normalized ResNet-50, (II) KNResNets significantly outperform the batch-independent competitors (the LayerNorm, GroupNorm, and LocalContextNorm based models) in terms of all considered performance metrics. Surprisingly, ResNet-50 based models perform worse than ResNet-34 counterparts for the competitors possibly because of the smaller kernel size they employ in ResNet-50 compared to ResNet-34 (1×1 instead of 3×3).

### 4.4 Differentially private image classification on ImageNet32×32

**Dataset.** ImageNet32×32 is the down-sampled version of ImageNet, where all images are resized to 32×32. For preprocessing, the feature values are divided by 255 for KNResNet-18, while they are normalized by the mean and SD of ImageNet for the layer and group normalized ResNet-18.

**Training.** We train KNResNet-18 as well as the GroupNorm and LayerNorm counterparts for 100 epochs using the SGD optimizer with zero-momentum and zero-weight decay, where the learning rate is decayed by factor of 2 at epochs 70, and 90. Note that BatchNorm is inapplicable to differential privacy. All models use the Mish activation (Misra, 2019). For parameter tuning, we consider learning rate values of {2.0, 3.0, 4.0}, clipping values of {1.0, 2.0}, and batch sizes of {2048, 4096, 8192}. We observe that learning rate of 4.0, clipping value of 2.0, and batch size of 8192 achieve the best performance for all models. Our differentially private training is based on DP-SGD (Abadi et al., 2016) from the Opacus library (Yousefpour et al., 2021) with $\varepsilon$=8.0 and $\delta$=8×10$^{-7}$. The privacy accountant is RDP (Mironov, 2017)

Table 4: Differentially private image classification on **ImageNet32×32**.

| Model | Normalization | Parameters | Top-1 accuracy |
|---|---|---|---|
| ResNet-18-BN | BatchNorm | 11.682 M | NA |
| ResNet-18-LN | LayerNorm | 11.682 M | 20.81 |
| ResNet-18-GN | GroupNorm | 11.682 M | 20.99 |
| KNResNet-18 (ours) | KernelNorm | 11.678 M | **22.01** |

**Results.** Table 4 lists the Top-1 accuracy values on ImageNet32×32 for different models trained in the aforementioned differentially private learning setting. As can be seen in the table, KNResNet-18 achieves significantly higher accuracy than the layer and group normalized ResNet-18.

## 5 Discussion

KNResNets incorporate only batch-independent layers such as the proposed KernelNorm and KNConv layers into their architectures. Thus, they perform well with very small batch sizes (Table 1) and are applicable to differentially private learning (Table 4), which are not the case for the batch normalized models. Unlike the batch-independent competitors such as LayerNorm, GroupNorm, and LocalContextNorm based ResNets, KNResNets provide higher or very competitive performance compared to the batch normalized counterparts in image classification and semantic segmentation (Tables 1-3). Moreover, KNResNets converge faster than the batch, layer, and group normalized ResNets in non-private and differentially private image classification as shown in Figure 4. These results verify our key claim: the kernel normalized models combine the performance benefit of the batch normalized counterparts with the batch-independence advantage of the layer, group, and local-context normalized competitors.

The key property of kernel normalization is the overlapping normalization units, which allows for kernel normalized models to *extensively* take advantage of the spatial correlation among the elements during normalization. Additionally, it enables KernelNorm to be combined with the convolutional layer effectively as a single KNConv layer (Equation 5 and Algorithm 1). The other normalization layers lack this property. BatchNorm, LayerNorm, and GroupNorm are global normalization layers, which completely ignore the spatial correlation of the elements. LocalContextNorm *partially* considers the spatial correlation dur-

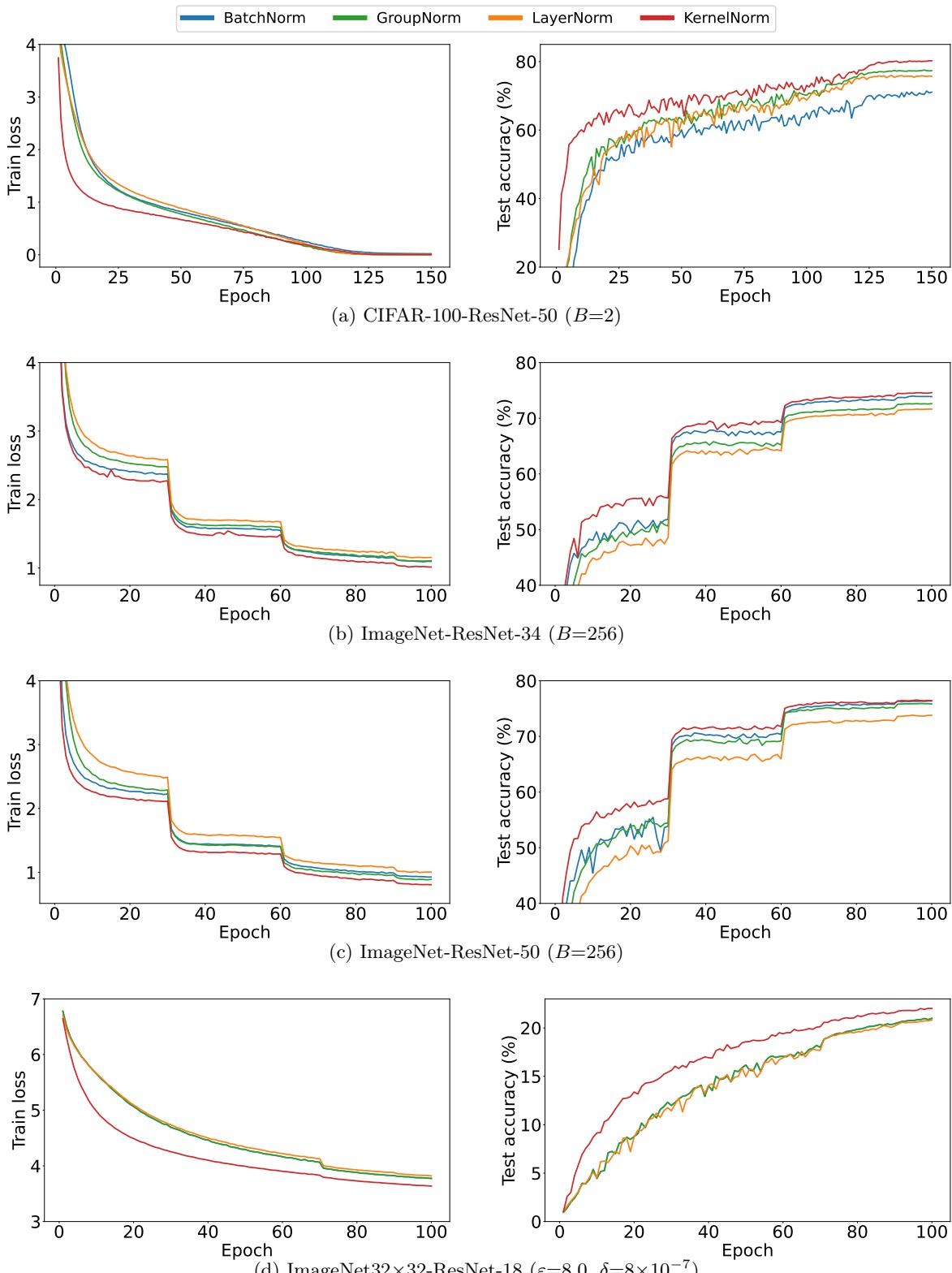

Figure 4: **Convergence rate** of the models for different case studies: Kernel normalized models converge faster than the competitors. Notice that BatchNorm is inapplicable to differential privacy; B: batch size.

ing normalization because it has no overlapping normalization units, and must use very large window sizes to achieve practical computational efficiency. Our evaluations illustrate that this characteristic of kernel normalization lead to significant improvement in convergence rate and accuracy achieved by KNResNets.

Normalizing the feature values of the input images using the mean and SD of the whole dataset is a popular data preprocessing technique, which enhances the performance of the existing CNNs due to feeding the normalized values into the first convolutional layer. This is unnecessary for KNConvNets because all KNConv layers including the first one are self-normalizing (they normalize the input first, and then, compute the convolution). This makes the data preprocessing simpler during training of KNConvNets.

Compared to the corresponding non-normalized networks, the accuracy gain in KNResNets originates from normalization using KernelNorm and regularization effect of dropout. To investigate the contribution of each factor to the accuracy gain, we train KNResNet-50 on CIFAR-100 with batch size of 32 in three cases: (I) without KernelNorm, (II) with KernelNorm and without dropout, (III) with KernelNorm and dropout. The models achieve accuracy values of 71.48%, 78.32%, and 80.18% in (I), (II), and (III), respectively. Given that, normalization using KernelNorm provides accuracy gain of around 7.0% compared to the non-normalized model. Regularization effect of dropout delivers additional accuracy gain of about 2.0%.

Prior studies show that normalization layers can reduce the sharpness of the loss landscape, improving the generalization of the model (Lyu et al., 2022; Keskar et al., 2016). Given that, we train LayerNorm, GroupNorm, and BatchNorm based ResNet-18 as well as KNResNet-18 on CIFAR-10 to compare the generalization ability and loss landscape of different normalization methods (experimental details in Appendix C). The layer, group, batch, and kernel normalized models achieve test accuracy of 90.32%, 90.58%, 92.11%, 93.27%, respectively. Figure 7 (Appendix C) visualizes the loss landscape for different normalization layers. According to the figure, KNResNet-18 provides flatter loss landscape compared to batch normalized ResNet-18, which in turn, has smoother loss landscape than the group and layer normalized counterparts. These results indeed indicate that KNResNet-18 and BatchNorm-based ResNet-18 with flatter loss landscapes provide higher generalizability (test accuracy) than LayerNorm/GroupNorm based ResNet-18.

There is a prior work known as convolutional normalization (ConvNorm) (Liu et al., 2021), which takes into account the convolutional structure during normalization similar to this study. ConvNorm performs normalization on the kernel weights of the convolutional layer (weight normalization). Our proposed layers, on the other hand, normalize the input tensor (input normalization). In terms of performance on ImageNet, the accuracy of KNResNet-18 is higher than the accuracy of the ConvNorm+BatchNorm based ResNet-18 reported in Liu et al. (2021) (71.17% vs. 70.34%).

We explore the effectiveness of KernelNorm on the *ConvNext* architecture (Liu et al., 2022) in addition to ResNets. ConvNext is a convolutional architecture, but it is heavily inspired by vision transformers (Dosovitskiy et al., 2020), where it uses linear (fully-connected) layers extensively and employs LayerNorm as the normalization layer instead of BatchNorm. To draw the comparison, we train the original *ConvNextTiny* model from PyTorch and the corresponding kernel normalized version (both with around 28.5m parameters) on ImageNet using the training recipe and code from TorchVision (2023b) (more experimental details in Appendix B). The original model, which is based on LayerNorm, provides accuracy of 80.87%. The kernel normalized counterpart, on the other hand, achieves accuracy of 81.25%, which is 0.38% higher than the baseline. Given that, KernelNorm-based models are efficient not only with ResNets, but also with more recent architectures such as ConvNext, which incorporates several architectural elements from vision transformers into convolutional networks.

We also make a comparison between KNResNets and the BatchNorm-based counterparts from the computational efficiency and memory usage perspectives (Tables 6 and 7 in Appendix D). For the batch normalized models, we employ two different implementations of the BatchNorm layer: The CUDA implementation (PyTorch, 2023a) and the custom implementation (D2L, 2023) using primitives provided by PyTorch. Because the underlying layers of KNResNets (i.e. KernelNorm and KNConv) are implemented using primitives from PyTorch, we directly compare KNResNets with ResNets based on the latter implementation of BatchNorm to have a fair comparison. According to Table 6, KNResNet-50 (our largest model) is only slower than batch normalized ResNet-50 by factor of 1.66. This slowdown is acceptable given the fact that KernelNorm is a local normalization layer with much more normalization units than BatchNorm as a global normalization

layer (Figure 1). The CUDA-based implementation of BatchNorm, moreover, is faster than that based on primitives from PyTorch by factor of 1.8. We can expect a similar speedup for KNResNets if the underlying layers are implemented in CUDA. Additionally, the memory usage of KNResNets is higher than the Batch-Norm counterparts as expected, which relates to the current implementation of the KNConv layer (more details in Appendix D). Notice that the most efficient implementation of KNResNets is not the focus of this study, and is left as a future line of improvement. Our current implementation, however, provides enough efficiency that allows for training KNResNet-18/34/50 on large datasets such as ImageNet.

## 6 Conclusion and Future Work

BatchNorm considerably enhances the model convergence rate and accuracy, but it delivers poor performance with small batch sizes. Moreover, it is unsuitable for differentially private learning due to its dependence on the batch statistics. To address these challenges, we propose two novel batch-independent layers called KernelNorm and KNConv, and employ them as the main building blocks for KNConvNets, and the corresponding residual networks referred to as KNResNets. Through extensive experimentation, we show KNResNets deliver higher or very competitive accuracy compared to BatchNorm counterparts in image classification and semantic segmentation. Furthermore, they consistently outperform the batch-independent counterparts such as LayerNorm, GroupNorm, and LocalContextNorm in non-private and differentially private learning settings. To our knowledge, our work is the first to combine the batch-independence of LayerNorm/GroupNorm/LocalContextNorm with the performance advantage of BatchNorm in the context of convolutional networks.

The performance investigation of KNResNets for object detection, designing KNConvNets corresponding to other popular architectures such as DenseNets (Huang et al., 2017a), and optimized implementations of KernelNorm and KNResNets in CUDA are promising directions for future studies.

## Acknowledgement

We would like to thank *Javad TorkzadehMahani* for assisting with the implementations and helpful discussions on the computationally-efficient version of the kernel normalized convolutional layer. We would also like to thank *Sameer Ambekar* for his helpful suggestion regarding fairer comparison among the normalization layers from the computational efficiency perspective.

This project was funded by the German Ministry of Education and Research as part of the PrivateAIM Project, by the Bavarian State Ministry for Science and the Arts, and by the Medical Informatics Initiative. The authors of this work take full responsibility for its content.

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

## A  KNResNet Architectures

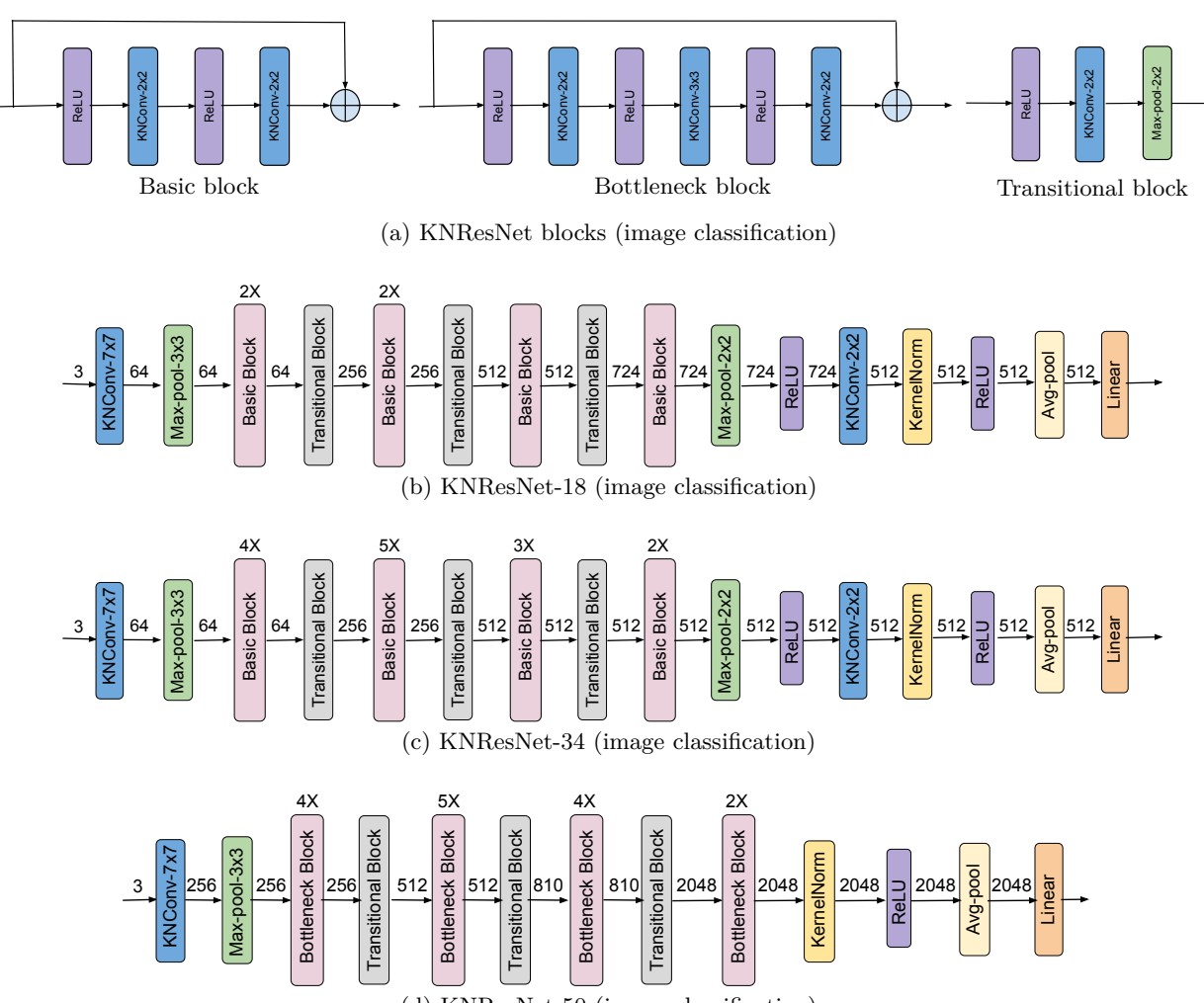

(a) KNResNet blocks (image classification)

(b) KNResNet-18 (image classification)

(c) KNResNet-34 (image classification)

(d) KNResNet-50 (image classification)

Figure 5: **KNResNets** for **image classification**: The dropout probability of the KNConv and KernelNorm layers are 0.05 and 0.25, respectively. For low-resolution images (e.g. CIFAR-100 with image shape of 32×32), the first KNConv layer is replaced by a KNConv layer with kernel size 3×3, stride 1×1, and padding 1×1, and the following max-pooling layer is removed. The *kX* (*k*=2/3/4/5) notation above the blocks means k blocks of that type. The numbers above arrows indicate the number of input/output channels of the first/last KNConv layer in the block. For KNResNet-18, the number of the output channels of the first KNConv layer (or the number of input channels of the second KNConv layer) is 256, 256, 512, and 724 for the first, second, third, and fourth set of basic blocks, respectively. For KNResNet-34, it is 256, 320, 640, and 843. For KNResNet-50, the number of output channels of the first and second KNConv layers are 64, 128, 201, and 512 in the first, second, third, and fourth set of bottleneck blocks, respectively. In KNResNet-50, the last transitional block and the last set of residual blocks use KNConv1×1 instead of KNConv2×2 to keep the number of parameters comparable to the original ResNet-50.

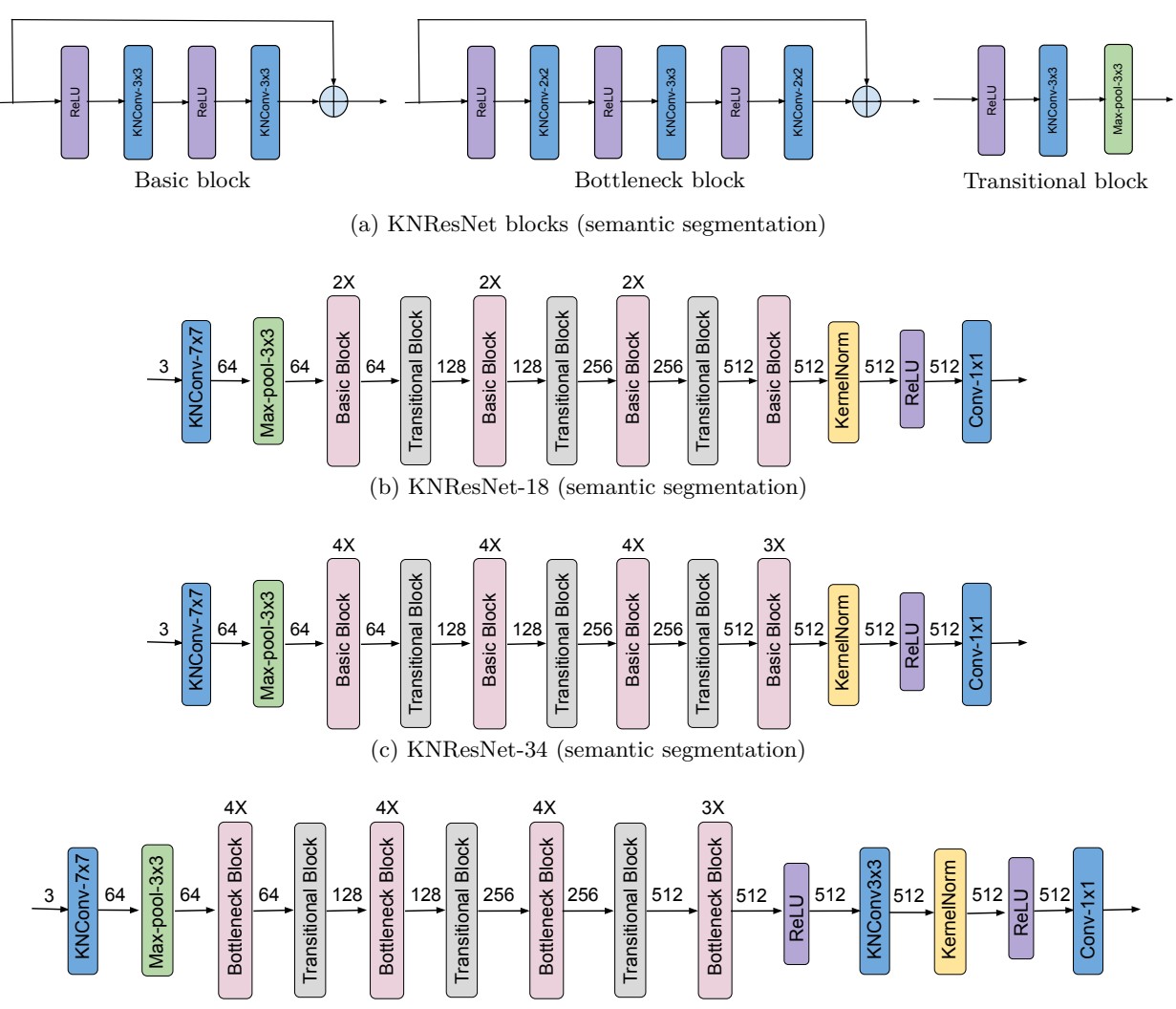

(a) KNResNet blocks (semantic segmentation)

(b) KNResNet-18 (semantic segmentation)

(c) KNResNet-34 (semantic segmentation)

(d) KNResNet-50 (semantic segmentation)

Figure 6: **KNResNets** for **semantic segmentation**: The dropout probability of the KNConv and Kernel-Norm layers are 0.1 and 0.5, respectively. For KNResNet-18, the number of the output channels of the first KNConv layer (or the number of input channels of the second KNConv layer) is 128, 256, 512, and 625 for the first, second, third, and fourth set of basic blocks. For KNResNet-34, they are 128, 256, 256, and 512, respectively. For KNResNet-50, the number of input/output channels of the middle KNConv layer are 128, 256, 458, and 512 for the first, second, third, and fourth set of bottleneck blocks. Unlike their counterparts for image classification, the KNConv and max-pooling layers in basic and transitional blocks employ kernel size of 3×3 instead of 2×2.

## B  Reproducibility

Table 5: Learning rate values achieving the highest accuracy on CIFAR-100.

| Model | Normalization | B=2 | B=32 | B=256 |
|---|---|---|---|---|
| ResNet-18-LN | LayerNorm | 0.0015625 | 0.0125 | 0.05 |
| PreactResNet-18-LN | LayerNorm | 0.0015625 | 0.0125 | 0.05 |
| ResNet-18-GN | GroupNorm | 0.0015625 | 0.025 | 0.1 |
| PreactResNet-18-GN | GroupNorm | 0.0015625 | 0.025 | 0.1 |
| ResNet-18-BN | BatchNorm | 0.00078125 | 0.025 | 0.2 |
| PreactResNet-18-BN | BatchNorm | 0.00078125 | 0.025 | 0.2 |
| KNResNet-18 | KernelNorm | 0.0015625 | 0.05 | 0.2 |
| ResNet-34-LN | LayerNorm | 0.0015625 | 0.0125 | 0.05 |
| PreactResNet-34-LN | LayerNorm | 0.0015625 | 0.0125 | 0.05 |
| ResNet-34-GN | GroupNorm | 0.0015625 | 0.025 | 0.1 |
| PreactResNet-34-GN | GroupNorm | 0.0015625 | 0.025 | 0.1 |
| ResNet-34-BN | BatchNorm | 0.00078125 | 0.025 | 0.1 |
| PreactResNet-34-BN | BatchNorm | 0.000390625 | 0.025 | 0.2 |
| KNResNet-34 | KernelNorm | 0.0015625 | 0.05 | 0.2 |
| ResNet-50-LN | LayerNorm | 0.00078125 | 0.0125 | 0.05 |
| PreactResNet-50-LN | LayerNorm | 0.0015625 | 0.0125 | 0.05 |
| ResNet-50-GN | GroupNorm | 0.00078125 | 0.0125 | 0.05 |
| PreactResNet-50-GN | GroupNorm | 0.0015625 | 0.025 | 0.1 |
| ResNet-50-BN | BatchNorm | 0.000390626 | 0.0125 | 0.1 |
| PreactResNet-50-BN | BatchNorm | 0.000195313 | 0.0125 | 0.2 |
| KNResNet-50 | KernelNorm | 0.0015625 | 0.025 | 0.2 |

**ConvNext on ImageNet**: To train the LayerNorm and KernelNorm based ConvNextTiny models on ImageNet, we employ the code and recipe from TorchVision (2023b), where the models are trained with total batch size of 1024 using the AdamW optimizer, learning rate of 0.001, and cosine learning rate scheduler for 600 epochs. Note that we use 4 GPUs with batch size of 256 per GPU rather than 8 GPUs with batch size of 128 per GPU in the original recipe due to the resource limitation.

## C   Loss Landscape

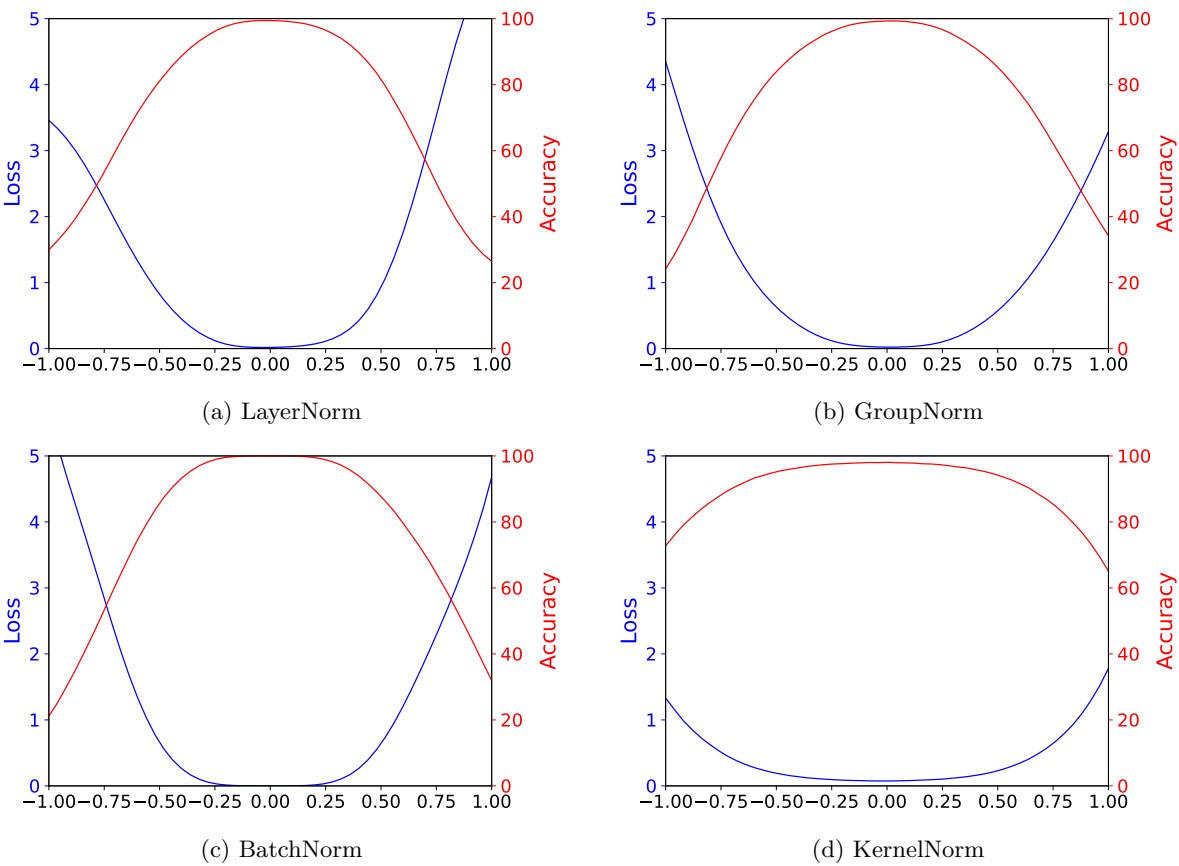

(a) LayerNorm

(b) GroupNorm

(c) BatchNorm

(d) KernelNorm

Figure 7: **Loss landscape** of different normalization layers: Kernel normalized ResNet-18 has flatter loss landscape compared to the batch, group, and layer normalized counterparts on CIFAR-10.

**ResNet-18 on CIFAR-10**: To compare the generalization ability and loss landscape of different normalization layers, we train BatchNorm, GroupNorm, LayerNorm, and KernelNorm based ResNet-18 on CIFAR-10. All models are trained for 70 epochs using batch size of 128 and tuned over learning rate values of $\{0.05, 0.1\}$. The weight decay is zero. The optimal learning rate is $0.05/0.05/0.1/0.1$ for layer/group/batch/kernel normalized ResNet-18. The preprocessing and augmentation scheme and the other training settings are the same as the CIFAR-100 experiments in Section 4. We employ the source code from Li et al. (2018a;b) to visualize the loss landscape in Figure 7.

# D   Running Time and Memory Usage

Table 6: **Training and inference** time per epoch for ImageNet: The experiments are conducted with 8 NVIDIA A40 GPUs with batch size of 32 per GPU; m: minutes, s: seconds.

| Model | Normalization | Implementation | Training time | Inference time |
|-------|--------------|----------------|---------------|----------------|
| ResNet-50-BN | BatchNorm | CUDA | 13m 23s | 6s |
| ResNet-50-BN | BatchNorm | Primitives from PyTorch | 23m 49s | 10s |
| KNResNet-50 (ours) | KernelNorm | Primitives from PyTorch | 39m 33s | 19s |
| ResNet-34-BN | BatchNorm | CUDA | 9m 12s | 5s |
| ResNet-34-BN | BatchNorm | Primitives from PyTorch | 12m 46s | 5s |
| KNResNet-34 (ours) | KernelNorm | Primitives from PyTorch | 27m 15s | 12s |
| ResNet-18-BN | BatchNorm | CUDA | 5m 28s | 4s |
| ResNet-18-BN | BatchNorm | Primitives from PyTorch | 7m 46s | 4s |
| KNResNet-18 (ours) | KernelNorm | Primitives from PyTorch | 13m 58s | 7s |

Table 7: **Memory usage** on ImageNet: The experiments are conducted with a single NVIDIA RTX A6000 GPU with batch size of 32; GB: Gigabytes.

| Model | Normalization | Implementation | Memory usage (GB) |
|-------|--------------|----------------|-------------------|
| ResNet-50-BN | BatchNorm | CUDA | 5.7 |
| ResNet-50-BN | BatchNorm | Primitives from PyTorch | 8.2 |
| KNResNet-50 (ours) | KernelNorm | Primitives from PyTorch | 13.6 |
| ResNet-34-BN | BatchNorm | CUDA | 3.6 |
| ResNet-34-BN | BatchNorm | Primitives from PyTorch | 4.4 |
| KNResNet-34 (ours) | KernelNorm | Primitives from PyTorch | 9.4 |
| ResNet-18-BN | BatchNorm | CUDA | 3.2 |
| ResNet-18-BN | BatchNorm | Primitives from PyTorch | 3.7 |
| KNResNet-18 (ours) | KernelNorm | Primitives from PyTorch | 7.2 |

The memory usage of KNResNets is higher than the BatchNorm counterparts. This observation is related to the current implementation of the KNConv layer, where the unfolding operation is performed in the kn_mean_var function (Algorithm 1) to compute the mean and variance of the units. We implemented KNConv in this fashion to avoid changing the CUDA implementation of the convolutional layer, which requires a huge engineering and implementation effort, and is outside the scope of our expertise.

In a hypothetical implementation of KNConv in CUDA, it would be possible to compute the mean/variance of the units directly inside the convolutional layer, and completely remove the kn_mean_var function, leading to substantially reducing the memory usage. This is because the units to compute convolution and mean/variance are the same, and those units are already available in the convolutional layer implementation.

Table 8: **Inference time | memory usage** for different stride, width (W) and height (H) values. The experiments are carried out with a single NVIDIA RTX A6000 GPU using batch size of 256 on the test set of CIFAR-100. The model contains four KNConv layers with kernel size of 3×3 and 256 channels; s: seconds, GB: Gigabytes.

| | W/H=32×32 | W/H=64×64 | W/H=128×128 |
|---|-----------|-----------|-------------|
| Stride=1×1 | 2.44s \| 2.80GB | 8.43s \| 4.94GB | 33.45s \| 13.44GB |
| Stride=2×2 | 0.64s \| 2.24GB | 1.07s \| 2.72GB | 2.91s \| 4.59GB |
| Stride=3×3 | 0.58s \| 2.16GB | 0.79s \| 2.40GB | 1.71s \| 3.28GB |

