# OpenReview forum: "Kernel Normalized Convolutional Networks"
_TMLR — Accepted by TMLR_

### Review · Reviewer_w1bf · 2023-10-16

**Summary Of Contributions:**

The paper introduces Kernel Normalized Convolutional Networks (KNConvNets), which aim to overcome the limitations of BatchNorm in convolutional neural networks (CNNs). It proposes two batch-independent layers, KernelNorm and KNConv, which take into account spatial correlations during normalization. These layers are integrated into KNConvNets and corresponding residual networks (KNResNets) to achieve competitive or superior performance compared to BatchNorm in image classification and semantic segmentation, even with small batch sizes.

**Audience:**

Yes

**Claims And Evidence:**

Yes

**Requested Changes:**

Please address the points in the weakness section.

- Evaluation of normalized kernel on models beyond ResNet
- More rigorous elaboration on DP, KNResNet benefit over other DP methods, and extended evaluation of DP.

**Strengths And Weaknesses:**

## Strengths
- The introduced concept of Kernel Normalized ConvNets is novel to the best of my knowledge
- The presented results are convincing, where the proposed method outperforms the compared methods
- The paper is well-written and easy to follow

## Weaknesses
- The evaluation is somewhat limited. The authors mainly study ResNet architectures. Hence it is not clear how generally applicable the kernel normalization is or if the authors found a technique that boosts only ResNet architectures. I suggest that the authors demonstrate the superiority of kernel normalization on architectures like DenseNet, NasNet, MobileNet, EfficientNet, etc.
- The authors claim, that the proposed kernel normalization improves DP, however, the DP evaluation is very limited. The authors should elaborate on why the DP performance is relatively low, and how DP was evaluated. I also recommend aligning with a common DP evaluation strategy from the literature instead of simply stating the Top-1 accuracy.
- The training times of the proposed method are a possible bottleneck for practitioners.

---

> ### Author Response · Authors · 2023-12-19
> **Response to Reviewer w1bf**
>
> We thank the reviewer for the comments, and kindly invite the reviewer to check out the revised draft too.
>
> > **Evaluation of normalized kernel on models beyond ResNet.**
>
> We chose ConvNext [1] as an architecture beyond ResNet to compare the performance of  KernelNorm with the baseline. The reason behind this choice was to simultaneously address the comment from this reviewer and Reviewer vx11, who stated that “comparing the proposed method with LayerNorm under ResNet architecture should not be fair.”
> ConvNext is a convolutional architecture, but it is heavily inspired by vision transformers, e.g. extensive use of linear (fully-connected) layers and employing LayerNorm as the normalization layer instead of BatchNorm.
>
> To make the comparison, we train the original ConvNext_Tiny model from TorchVision and its kernel normalized counterpart (both with ~28.5m parameters) on ImageNet using the training recipe and code from the PyTorch website [2] (more experimental details in Appendix B of the revised draft). The original model, which is based on LayerNorm, provides Top-1 accuracy of 80.87%. The kernel normalized ConvNext_Tiny, on the other hand, achieves Top1 accuracy of 81.25%, which is 0.38% higher than the baseline. Given that, KernelNorm based models are efficient not only with ResNet architectures, but also with more recent architectures such as ConvNext, which incorporates several architectural elements from vision transformers into convolutional networks.
>
> Note that due to the time constraint and limited resources we had as an academic institution, we could not spend much time on optimizing the ConvNext architecture for KernelNorm. That is, the accuracy gain from KernelNorm compared to LayerNorm for the ConvNext models could further be improved by tuning the architecture for KernelNorm.
>
> We also added a paragraph in the Discussion section of the revised draft to discuss the results on the ConvNext architecture.
>
> > **More rigorous elaboration on DP, KNResNet benefit over other DP methods, and extended evaluation of DP.**
>
> In this draft, we introduce the idea of kernel normalization, and show its effectiveness mainly in non-private training. We also provide a case study using ResNet-18 on ImageNet32x32 in DP to illustrate the great potential of KernelNorm for DP settings.
>
> In a separate paper entitled “Kernel Normalized Convolutional Networks for Privacy-preserving Machine Learning”, published in the IEEE SaTML conference (2023), we conducted thorough experiments on privacy related domains including federated learning (FL), differential privacy (DP), and differentially private federated learning (DP-FL) using VGG, DenseNet, and ResNet architectures. In the paper, we drew a detailed comparison between LayerNorm, GroupNorm, and KernelNorm, and show that KernelNorm-based models significantly outperform the competitors in terms of both accuracy and convergence rate (communication efficiency) in FL, DP, and DP-FL environments. We also improved the state-of-the-art accuracy on CIFAR-10 and ImageNette (a subset of ImageNet) using our proposed kernel normalized model under DP training.
>
> Note that all three reviewers unanimously accepted the paper, and agreed with us regarding the efficiency of kernel normalization for privacy-related applications.
>
> Surprisingly, our SaTML paper on the application of KernelNorm in privacy-related domains got accepted first. This is mainly because the ImageNet experiments provided in this submission  (Table 2) are very resource-intensive and time-consuming given the limited resources we have as an academic institution. Thus, it took a while for us to conduct and finalize the experiments on ImageNet. In the SaTML paper, following the previous studies, we focused on smaller datasets such as CIFAR-10/100 on FL, DP, and DP-FL.
>
> Notice that our SaTML paper cites this paper as the original paper for KernelNorm, and there is no overlap between this submission and the SaTML paper in terms of the conducted experiments and contributions.
>
> We uploaded the anonymized version of our SaTML paper as supplementary material.
>
> > **The authors should elaborate on why the DP performance is relatively low.**
>
> In general, there is a large gap between the performance of the model on non-private and DP training. This is because the gradient clipping and random noise injection in DP highly impacts the accuracy of the model. Given that, it is not surprising we report accuracy around 21.0-22.0% for ImageNet32x32 using ResNet-18 under DP training with epsilon=8.0. The study by Kurakin et al. [3] (Table 4 on Page 8) reports even lower accuracy, 6.9% with epsilon=13.2, on ResNet-18-ImageNet compared to our study.
>
> 1) Liu et al., “A convnet for the 2020s”, https://arxiv.org/abs/2201.03545, CVPR 2022.
> 2) https://github.com/pytorch/vision/tree/main/references/classification#convnext
> 3) Kurakin et al., “Toward Training at ImageNet Scale with Differential Privacy”, https://arxiv.org/abs/2201.12328 , 2022.

---

### Review · Reviewer_vx11 · 2023-11-22

**Summary Of Contributions:**

The paper tackles the batch-dependent limitation of Batch normalization (the model with BN layers could crash when learned on small batch size, and is not suitable under differentially private training) by introducing a new local, batch-independent normalization (KernelNorm). Authors further design kernel-normalized convolutional networks (KNConvNets), which are based on KernelNorm and can achieve competitive or higher performance of BN and other normalization techniques (LayerNorm, GroupNorm) under different settings.

**Audience:**

Yes

**Claims And Evidence:**

Yes

**Requested Changes:**

Please refer to my concerns

**Strengths And Weaknesses:**

This paper is easy to follow, authors first point out the limitation of BatchNorm, then they propose a new normalization method, which can overcome the drawback of BN under the setting that BN fails. The empirical experiment verifies their proposed method, which can tackle the problem of BN when combat with small batch size and under differentially private training. Besides, the computation cost of the suggested method is well estimated, with acceptable running time (and could be further improved as the author discusses).

Here are some of my concerns that relate to the paper:
1. Author indicates that overlap normalization can "extensively take into account the spatial correlation among the elements". In traditional CNN, the convolutional layers (Convs) already have this trait, so what is the advantage of overlap normalization in the proposed method over Convs in this case?
2. LayerNorm achieves better performance in ViT architecture, therefore comparing the proposed method with LayerNorm under ResNet architecture should not be fair.
3. Generalization ability: The normalization layer, besides helping the model learn more stable and achieve better performances, also indicates that helps improve the generalization of the learned model [1]. So it will be better if the authors can propose some compared results about the generalization performance of their methods with other normalization techniques (for example, the generalization could be clarified through visualizing the loss landscape of the learned model).

**Reference**

[1] Understanding the Generalization Benefit of Normalization Layers: Sharpness Reduction. NeurIPS 2022

---

> ### Author Response · Authors · 2023-12-19
> **Response to Reviewer vx11**
>
> We thank the reviewer for the comments, and kindly invite the reviewer to check out the revised draft too.
>
> > **Author indicates that overlap normalization can "extensively take into account the spatial correlation among the elements". In traditional CNN, the convolutional layers (Convs) already have this trait, so what is the advantage of overlap normalization in the proposed method over Convs in this case?**
>
> The complete sentence is “extensively take into account the spatial correlation among the elements during normalization”. The phrase “during normalization” is key here. The convolutional layer has this characteristic similar to KernelNorm, but during computation of convolution. The main advantage of overlapping normalization units in KernelNorm is that it can effectively be combined with the convolutional layer, which indeed has overlapping convolutional units, forming the KNConv layer. That is, the overlapping normalization units from KernelNorm and the overlapping convolutional units from the convolutional layer are the same in KNConv layer. In summary, KernelNorm has the aforementioned characteristic during normalization, the convolutional layer has that property during convolution computation, and the KNConv layer has the trait during both normalization and convolution computation.
>
> Note that KernelNorm is the only normalization layer that can normalize exactly the same elements used to compute the convolution thanks to its overlapping units. Considering the spatial correlation among the elements during both normalization and computation of convolution, which is the case for KNConv, provides considerable accuracy gain as our experiments show.
>
> > **LayerNorm achieves better performance in ViT architecture, therefore comparing the proposed method with LayerNorm under ResNet architecture should not be fair.**
>
> We compare the performance of KernelNorm with LayerNorm on the ConvNext architecture [1]. ConvNext is a convolutional architecture, but it is heavily inspired by vision transformers, e.g. extensive use of linear (fully-connected) layers and employing LayerNorm as the normalization layer instead of BatchNorm.
> To make the comparison, we train the original ConvNext_Tiny model from the TorchVision library and its kernel normalized counterpart (both with ~28.5m parameters) on ImageNet using the training recipe and code from the PyTorch website [2] (more experimental details in Appendix B of the revised draft). The original model, which is based on LayerNorm, provides Top-1 accuracy of 80.87%. Kernel normalized ConvNext_Tiny, on the other hand, achieves Top1 accuracy of 81.25%, which is 0.38% higher than the baseline. Given that, KernelNorm outperforms LayerNorm not only on ResNets, but also on ConvNext, which is designed and optimized based on LayerNorm.
>
> It is worth mentioning that LayerNorm can be considered as a special case of KernelNorm, where the kernel size is (width, height) of the input tensor and dropout probability is zero. Thus, KernelNorm is indeed applicable to any architecture whose underlying normalization layer is LayerNorm such as vision transformers.
>
> Note that due to the time constraint and the limited resources we had as an academic institution, we could not spend much time on optimizing the ConvNext architecture for KernelNorm. That is, the accuracy gain from KernelNorm compared to LayerNorm for the ConvNext models could further be improved by tuning the architecture for KernelNorm.
>
> We also added a paragraph in the Discussion section to discuss the results on the ConvNext architecture.
>
> > **Generalization ability and loss landscape**
>
> To compare the generalization ability and loss landscape of different normalization layers, we train BatchNorm, GroupNorm, LayerNorm, and KernelNorm based ResNet-18 on CIFAR-10. The layer, group, batch, and kernel normalized models achieve test accuracy of 90.32%, 90.58%, 92.11%, 93.27%, respectively. Now, we visualize the loss  landscape using the code provided in [3, 4] for different normalization layers (Figure 7 in  Appendix C). According to the figure, KNResNet-18 provides flatter loss landscape compared to batch normalized ResNet-18, which in turn, has smoother loss landscape than the group and layer normalized counterparts. These results indicate that KNResNet-18 and BatchNorm-based ResNet-18 with flatter loss landscapes provide higher generalizability (test accuracy) than LayerNorm/GroupNorm-based ResNet-18.
>
> We also added a paragraph in the Discussion section in this regard.
>
> Note that we chose CIFAR-10 as the dataset because the code from [4] supports only CIFAR-10.
>
> 1) Liu et al., “A convnet for the 2020s”, https://arxiv.org/abs/2201.03545, CVPR 2022.
> 2) https://github.com/pytorch/vision/tree/main/references/classification#convnext
> 3) Li et al., "Visualizing the loss landscape of neural nets", https://arxiv.org/abs/1712.09913, Advances in neural information processing systems, 2018.
> 4) https://github.com/tomgoldstein/loss-landscape

---

### Review · Reviewer_awgw · 2023-11-22

**Summary Of Contributions:**

This paper proposes a new normalization method for Convolutional Neural Networks (ConvNets) termed 'kernel normalization.' This method calculates the mean and variance across a spatial region determined by the convolutional kernel's width, height, and input channels. The authors present empirical verifications in image classification, semantic segmentation, and differential privacy. They demonstrate that classification performance remains unaffected by a reduced training batch size, while performance in the other two tasks shows improvement.

**Audience:**

Yes

**Broader Impact Concerns:**

Not Applicable.

**Claims And Evidence:**

No

**Requested Changes:**

Requested Changes:

1. Ablation study on the usage of dropout layer.

2. The extra memory / runtime for the proposed method.

3. Extra memory / inference speed as a function of kernel stride / width / height.

And the inline citation format is wrong. The authors might mix the usage of \citet and \citep.

Missing related work on weight normalization:

[1] Can We Gain More from Orthogonality Regularizations in Training Deep CNNs?, NIPS 2018.

[2] Orthogonal Convolutional Neural Networks, CVPR 2020.

[3] Deep Isometric Learning for Visual Recognition, ICML 2020.

**Strengths And Weaknesses:**

(+) This is definitely an interesting operator that can potentially replace layernorm or groupnorm in ConvNets. It aligns more closely with the design of convolutional layers and leverages the spatial correlation inherent in the network.

(+) The writing is clear, straightforward, and easy to understand.

However, the main weakness lies in the design of the experiments. From my understanding, this manuscript only presents the performance of the proposed method alongside previous ones. While this effectively showcases the capabilities of the proposed method, the information provided is not particularly insightful. More specifically:

(-) The manuscript lacks discussion on the use of dropout. Unlike other input normalizations, this method employs a dropout layer before computing the means. While this approach is logical, there are no experiments demonstrating its impact on performance.

In the experiments, it appears that the method performs better with smaller datasets like CIFAR and in the early stages of training (as seen in Figure 4). This outcome is often observed with larger regularization factors (such as dropout, higher weight decay). Hence, it's challenging to determine if dropout plays a more significant role than expected in these experiments.

(-) Practically speaking, the training time is excessively long. It is standard practice to implement the proposed operator efficiently; failing to do so could substantially diminish the paper's contribution.

(-) There is no discussion on the additional memory usage. The Kernel norm layer effectively duplicates overlapping regions of an input tensor, which should increase GPU memory consumption. The authors should provide a comparison of GPU memory usage for different kernel shapes. When the stride is small and the kernel size is large, the additional memory requirement could be even more significant. This aspect is crucial as it also influences how researchers might design the kernel shape when utilizing the proposed method.

(-) Similarly, a discussion on the running time (at test time) is necessary. By the same logic that affects memory usage, the impact on running time should also be evaluated and discussed.

---

> ### Author Response · Authors · 2023-12-19
> **Response to Reviewer awgw**
>
> We thank the reviewer for the comments, and kindly invite the reviewer to check out the revised draft too.
>
> > **Ablation study on the usage of dropout layer**
>
> We kindly ask the reviewer to check the initial draft, where we already provided the ablation study on the usage of dropout in the Discussion section:
>
> “Compared to the corresponding non-normalized networks, the accuracy gain in KNResNets originates from normalization using KernelNorm and regularization effect of dropout. To investigate the contribution of each factor to the accuracy gain, we train KNResNet-50 on CIFAR-100 with batch size of 32 in three cases: (I) without KernelNorm, (II) with KernelNorm and without dropout, (III) with KernelNorm and dropout. The models achieve accuracy values of 71.48%, 78.32%, and 80.18% in (I), (II), and (III), respectively. Given that, normalization using KernelNorm provides accuracy gain of around 7.0% compared to the non-normalized model. Regularization effect of dropout delivers additional accuracy gain of about 2.0%.”
>
> > **The extra memory / runtime for the proposed method**
>
> We added the inference time and memory usage to the corresponding tables in Appendix D. As expected, the memory usage of KNResNets is higher than the BatchNorm counterparts. This observation is related to the current implementation of the KNConv layer, where the unfolding operation is performed in the kn_mean_var function (Algorithm 1) to compute the mean and variance of the units. We implemented KNConv in this fashion to avoid changing the CUDA implementation of the convolutional layer, which requires a huge engineering and implementation effort, and is outside the scope of our expertise.
>
> In a hypothetical implementation of KNConv in CUDA, it would be possible to compute the mean/variance of the units directly inside the convolutional layer, and completely remove  the kn_mean_var function, leading to substantially reducing the memory usage. This is because the units to compute convolution and mean/variance are the same, and those units are already available in the convolutional layer implementation.
>
> > **Extra memory / inference speed as a function of kernel stride / width / height**
>
> We added Table 8 to Appendix D, which lists the inference time and memory usage for different stride and width/height values. As expected, lower stride values and higher width/height values lead to higher inference time and memory usage.
>
> > **In the experiments, it appears that the method performs better with smaller datasets like CIFAR and in the early stages of training (as seen in Figure 4).**
>
> Figure 4a corresponding to CIFAR-100 shows the convergence rate of the methods with small batch size of 2, where BatchNorm performs poorly. Figures 4b and 4c, on the other hand, plot the convergence rate on ImageNet with batch size of 256, where BatchNorm performs well. In other words, the difference between the performance of the methods are more related to the batch size than the dataset.
>
> > **It is standard practice to implement the proposed operator efficiently; failing to do so could substantially diminish the paper's contribution.**
>
> We make at least three main contributions in this paper: (1) A novel normalization method and the corresponding KernelNorm and KNConv layers, (2) designing KNResNets as batch-independent models using the proposed layers, and (3) extensive experiments in image classification and semantic segmentation to show the effectiveness of our method. The optimal implementation of the method in CUDA could be an additional contribution of this paper, which we left as a future work, due to the huge implementation effort it requires. Given that, we kindly disagree with the reviewer that one missing additional contribution considerably diminishes the other contributions of this study!
>
> > **Missing related work on weight normalization**
>
> We cited the studies related to weight normalization mentioned by the reviewer in the revised draft.

---

### Decision · Action_Editor_vvhY · 2024-01-23

**Recommendation:** Accept as is

**Comment:**

All in all, this is a well written piece of work that shows a novel and interesting way to do normalization in conv-nets. The authors show that it is competitive on a reasonable range of tasks and the disadvantages pointed out by the reviewers do not seem insurmountable given the potential benefit this method could bring (in cases where other normalization techniques like batch norm are not working too well).

**Audience:**

The work should be interesting to a rather wide audience, most folks training conv-nets (and such) would find this useful as a normalization strategy to consider.

**Claims And Evidence:**

This work presents an alternative normalization method for training conv-nets, called kernel normalization. The idea is to use the mean/variance across a spatial region (corresponding to the kernel dimensions). This kind of operator could potentially replace layer norm or group norm. The work presents evidence that shows this method being competitive with batch normalization and other normalization techniques in a variety of settings (classification, segmentation, and differential privacy).

While the method resolves some of the issues present in competing normalization methods (notably as it relates to small batch sizes), it does come with some trade-offs related to training time, memory usage and run-time inference cost. Some of the reviewers noted that the improvements on the whole could be more significant; the latter concern is somewhat mitigated by the fact that it’s increasingly harder to beat existing baselines over time.

The rebuttal presented a lot of evidence requested by the reviewers and on the whole made the claims stronger as a result.